# The Prevalence and Characteristics of Chronic Ankle Instability in Elite Athletes of Different Sports: A Cross-Sectional Study

**DOI:** 10.3390/jcm11247478

**Published:** 2022-12-16

**Authors:** Congda Zhang, Na Chen, Jingsong Wang, Zhengzheng Zhang, Chuan Jiang, Zhong Chen, Jianhui Fang, Juhua Peng, Weiping Li, Bin Song

**Affiliations:** 1Department of Orthopedics, Sun Yat-sen Memorial Hospital, Guangzhou 510130, China; 2Department of Rehabilitation, Guangdong Sport Hospital, Guangzhou 510105, China

**Keywords:** ankle sprains, chronic ankle instability, elite athletes, prevalence, sports

## Abstract

Background: Ankle sprains are one of the most common injuries in athletic populations. Misdiagnosed and untreated ankle sprains will cause chronic ankle instability (CAI), which can significantly affect the performance of athletes. This study aimed to investigate the prevalence and characteristics of CAI in elite athletes of different sports. Method: This cross-sectional study included 198 elite athletes from Guangdong provincial sports teams. All participants answered a questionnaire about ankle sprains and ankle instability. The severity of their ankle instability was evaluated by the Cumberland Ankle Instability Tool (CAIT). Participants further underwent clinical examinations from sports medicine doctors to determine the presence and characteristics of ankle instability. The datasets were analyzed to determine the differences in prevalence between age, gender, sports teams, and sports categories. Results: In 198 athletes, 39.4% (n = 78) had bilateral CAI while 25.3% (n = 50) had unilateral CAI. Female athletes had a higher prevalence of CAI than male athletes in the study (*p* = 0.01). Prevalence showed differences between sports categories, and were significantly higher in acrobatic athletes than non-contact athletes (*p* = 0.03). Conclusions: CAI was highly prevalent among elite athletes in this study, with female athletes and athletes in acrobatic sports being associated with a higher risk of developing CAI in their professional careers. Therefore, extra precautions need to be taken into account when applying ankle protections for these athletes.

## 1. Introduction

Ankle sprain is one of the most common musculoskeletal injuries, with a high incidence in competitive sports [1,2]. Acute ankle sprain has a high recurrence rate, which often leads to the development of chronic ankle instability (CAI). CAI is characterized by patients who (1) suffered their first ankle sprain over a year prior; (2) exhibit a propensity for recurrent ankle sprains; (3) experience frequent perceptions of the ankle giving way; and (4) suffer from repetitive pain, swelling, diminished ROM, weakness, and reduced daily function [3,4]. The reoccurring harm, restricted movement, and imbalance caused by CAI leads to further injuries, which significantly affect the athletes’ career expectation for practice and competition [5]. The high prevalence of re-injury and subsequent development of CAI also have long-term consequences such as increased risk of ankle joint degeneration, specifically ankle joint post-traumatic osteoarthritis [6,7,8].

In 2011, Hiller et al. [9] presented an extended model based on the Hertel model [10] for classifying patients with CAI including multiple clinical subgroups: mechanical instability, perceived instability, recurrent sprains, or combinations of these three conditions. However, it is still unclear which subgroups are causative, associative, or by-products of the condition, due to inadequate clinical data or validation research [11]. Lack of conclusive evidence makes these models unreliable when applying them to actual patients. To classify patients with CAI, a combination of self-reported questionnaires, clinical assessment, and objective measurements should be considered.

The International Ankle Consortium (IAC) published a diagnosis standard for CAI in 2014, including: a previous severe lateral ankle sprain (LAS), recurrent LAS, a feeling of the ankle “giving way”, and validated self-reported questionnaires [4]. Although the incidence of ankle sprains in athletes has been investigated in many studies [12,13,14], the presence of CAI was identified only by self-reported questionnaires, without using any past medical history (such as a significant LAS, recurrent LAS, and ankles “giving way”). It is difficult to compare the CAI prevalence among the studies because of the different criteria. To our knowledge, no study has previously compared the characteristics of instability in various elite athletes with CAI. Additionally, the physiological profiles of different sports are distinct, which might lead to different injury rates. It is essential to investigate the prevalence of CAI in different types of athletes according to the IAC criteria and their characteristics. Furthermore, these results could also help to make more proper adjustments for individuals’ training programs.

The purposes of the study were (1) to discover the positive rate of CAI in elite athletes from different sports based on the IAC criteria; (2) to compare the differences in the demographic and ankle characteristics between athletes with or without CAI; and (3) to investigate if the prevalence is impacted by the category of sport (acrobatic, contact, and non-contact). We hypothesized that the prevalence of CAI in elite athletes is influenced by sex, age, and the category of sport.

## 2. Materials and Methods

### 2.1. Study Design

This study represents a cross-sectional analysis of data on the CAI prevalence of elite athletes from Guangdong provincial sports team. This study was performed in the Guangdong Sport Hospital (Guangzhou, China) from December 2021 to March 2022. All participants signed informed consent forms before the study began.

Athletes from different sports were diagnosed with or without CAI using questionnaires and clinical assessments based on the criteria recommended by IAC [4]. All athletes filled in a questionnaire which included their demographic characteristics, history of ankle sprains, and episodes of ankle giving way or instability. Furthermore, another questionnaire (the Cumberland Ankle Instability Tool (CAIT)) recommended by IAC was used to present the participants’ self-reported ankle instability [15]. Each participant was also given a clinical assessment of their feet and ankles by experienced orthopedic surgeons from the sports medicine department. Inclusion criteria of this study were as follows: (1) participants were members of Guangdong provincial sports teams (including diving, basketball, weightlifting, badminton, track and field, fencing, *Sanda*, trampolining, and gymnastics) and (2) signed the informed consent form. The exclusion criteria included (1) acute injuries in the lower limbs (sprains or fractures); (2) a history of lower limb surgery or fracture; and (3) being unable to complete questionnaires or assessments.

### 2.2. Outcome Measures

Demographic characteristics of all participants (age, gender, body height, weight, and body mass index) and questionnaires for CAI were collected individually. The clinical assessments were measured separately for the right and left ankle. The diagnosis criteria of CAI included: (1) a history of significant ankle sprain (more than 12 months); (2) the injured ankle having a history of a feeling of “giving way” and/or recurrent sprain; and (3) a CAIT score ≤ 24. The clinical assessment to determine the severity of ankle instability was performed via anterior drawer tests, talar tilt tests, and other assessments.

Foot and ankle characteristics including the arch of the foot and dorsiflexion ranges were measured using the navicular drop test [16] and dorsiflexion lunge test [17]. For the navicular drop test, the level of the navicular bone was measured in two positions: standing (weight bearing) and relaxed sitting (non-weight bearing). A small rigid ruler was used to measure from the ground to the most prominent part of navicular tuberosity at the neutral talar position. The differences in the two measurements were recorded. For the dorsiflexion lunge test, athletes were asked to face the wall and squat maximally against their knee to the wall without lifting the heel. The distance from wall to the big toe was then recorded (toe–wall distance, TW). Both of these tests were repeated three times consecutively for both legs. All the tests were performed at the same place by the same person.

### 2.3. Statistical Analysis

Differences in demographics and ankles characteristics between athletes with CAI and without CAI were examined using the Mann–Whitney U test or independent T test. The Chi-square test was used to determine the difference in the presence of CAI between sports and gender. *p* < 0.05 was considered statistically significant and all the analyses were performed using SPSS software (version 21.0).

## 3. Results

Figure 1 displays the process for inclusion in this study. Initially, 227 elite athletes from Guangdong provincial sports teams were recruited for this study. Twenty-five athletes were excluded due to a history of lower limb surgery or lower limb fractures. Four athletes who did not complete the questionnaires or clinical assessments were also excluded. Athletes (n = 198) from nine different sports were included in this study (24 diving, 40 basketball, 19 weightlifting, 20 badminton, 20 track and field, 23 fencing, 12 *Sanda*, 20 trampolining, and 20 gymnastics).

A total of 170 athletes had previous ankle sprains, while 28 athletes (14.1%) had no history of ankle sprain. Of the 170 athletes with previous ankle sprains, 128 athletes met the criteria of having CAI. Table 1 illustrates the complete demographic data of all participating athletes. There were no significant demographic (age, height, weight, and body mass index) differences between athletes with CAI and without CAI. The CAIT value for athletes with CAI was significantly lower than for athletes without CAI (*p* < 0.05, Table 1). The prevalence of CAI was significantly higher in female than male athletes (X^2^(1) = 9.2, *p* = 0.010) (Table 2).

For foot and ankle characteristics, 206 ankles meet the IAC criteria (left: 100; right: 106); their means (SD) of navicular drop distance were 0.39 (0.17) cm and 0.41(0.19) cm for left and right, respectively, while for toe–wall distance they were 11.40 (3.00) cm and 12.01 (3.37) cm for left and right, respectively. In those 190 ankles which did not meet the CAI inclusion criteria (left: 98; right: 92), for navicular drop distance they were 0.42 (0.18) cm and 0.42 (0.19) cm for left and right, respectively, and for toe–wall distance they were 11.51 (3.50) cm and 11.94 (3.28) cm for left and right, respectively. There was no significant difference in the navicular drop tests between the ankles that did and did not meet the CAI inclusion criteria (left: *p* = 0.12; right: *p* = 0.68, Figure 2), or in the dorsiflexion ranges (toe–wall distance) (left: *p* = 0.82; right: *p* = 0.88, Figure 3).

Table 3 lists the number and percentage of athletes with ankle sprain history and CAI for different sports. Diving and trampolining had the highest percentage (75.0%), followed by gymnastics (70.0%) (Table 3). However, no sports team proved to have a higher prevalence than all the others. Athletes participating in gymnastics, trampolining, and diving were grouped as being in “acrobatic” sports. Basketball and *Sanda* were grouped as “contact” sports. Weightlifting, badminton, fencing, and track and field were grouped as “non-contact” sports. For these three categories, the prevalence of CAI was 73.4% for acrobatic (47/54), 67.3% for contact (35/52), and 56.1% for non-contact (46/82) sports (Table 4). Athletes in the acrobatic group had a significantly higher prevalence of CAI than athletes in the non-contact group (X^2^(1) = 4.674, *p* = 0.03).

Additionally, there was no significant differences in terms of the CAIT value in the 198 athletes among the nine sports teams (Figure 4). Table 5 lists the positive rate of anterior drawer tests and talar tilt tests for CAI athletes in different sports. A total of 73.3% (151/206) who met the IAC criteria had positive results on anterior drawer tests and 44.2% (91/206) on talar tilt tests.

## 4. Discussion

Our study investigated the prevalence of chronic ankle instability in 198 elite athletes from different sports, as well as the differences in demographic and ankle characteristics between athletes with CAI and without CAI. A total of 85.9% of all the elite athletes had previous ankle sprains, and the prevalence of CAI was 64.6% (128/198). The prevalence of CAI was significantly different among genders and sports categories. Female elite athletes had a significantly higher presence of CAI than male elite athletes. Between sports categories, CAI was significantly higher in “acrobatic” athletes than in “non-contact” athletes. However, there was no significant difference in the ankle characteristics between the ankles which did and did not meet the CAI inclusion criteria.

The primary aim of our study was to provide persuasive evidence of the prevalence and characteristics of CAI. Considering that most previous CAI articles investigated the general population, we aimed to discover whether there are CAI problems among elite athletes without focusing on identifying the risk factors of CAI. In our study, the prevalence of CAI was 64.6% (128/198) (Table 1). Previous studies reported that the prevalence of perceived ankle instability was between 20% and 47% [18,19,20]. The prevalence of CAI varies dramatically in specific groups of subjects. The high prevalence of CAI in elite athletes might be caused by the following factors: (1) the participants in this study were playing in a higher professional league and compete at a higher intensity, as well as having a higher rate of previous ankle injures; (2) elite athletes might place more hyperflexible pressure on their ankles than general athletes or amateur athletes, especially in acrobatic sports; and (3) most elite athletes started highly specialized training as children, which exposes them to a higher risk of overuse injury. Future study is required to fully investigate the relevance between the professional level of the athlete and the prevalence of CAI.

Our study investigated the prevalence of CAI in elite athletes based on the inclusion and exclusion criteria of the International Ankle Consortium (IAC). In several studies [19,20], perceived ankle instability was defined as CAI without providing any specific definition for the criteria they were using. Different inclusion and exclusion criteria applied in previous articles caused discrepancies in conclusions. Therefore, to investigate the prevalence of CAI, a standard method for identifying the origin of perceived ankle instability is needed. Previous studies [12,21] defined CAI only on the basis of self-reported questionnaires which assessed ankle instability. To improve the quality of study on CAI pathology, the International Ankle Consortium (IAC) provided recommendations for the criteria in 2014 [4]. Recent studies [22,23] investigating participants with or without CAI have often used the criteria based on the recommendations provided by the IAC.

Our study found that male athletes had a lower prevalence of CAI than female athletes (Table 2). Some explanations for the increased risk of ankle sprains among female athletes may include: different neuromuscular control [24], increased joint laxity [25], and decreased postural control [26]. Females have an increased tibial varum and calcaneal eversion range of motion, which may contribute to greater risk of ankle injury [27]. Similar to anterior cruciate ligament tears, which also have a higher prevalence in female athletes, decreased postural control may be a cause [28]. Future research could focus on whether the differences in incidence of ankle sprains between males and females are related to activity-specific behaviors, training behaviors, or anatomical and physiological factors.

In our study, there was no significant difference in the ankle characteristics (including the arch of foot and dorsiflexion ranges) between participants that did and did not meet the CAI inclusion criteria (Figure 2 and Figure 3). For the navicular drop test, a navicular drop > 1.0 cm was regarded as pes planus [16]. A previous study demonstrated that the navicular drop test is a reliable and simple test to evaluate the medial longitudinal arch height [29]. Ashraf et al. [30] reported that the prevalence of pes planus for 18- to 25-year-olds was 32.6%. However, none of the participants’ navicular drop exceeded 1.0 cm in this study. It is obvious that potential professionals tend to be selected from a young age. Langarika et al. [17] observed that the dorsiflexion lunge test has a high relevance with the reference standard for assessing dorsiflexion range of motion and the toe–wall distance is a simple method for evaluating this. Martin et al. found that reduced dorsiflexion may be associated with a higher risk of sprain and that ankle dorsiflexion would be altered after ankle injury [31]. Although no significant decrease in weight-bearing dorsiflexion was found in CAI athletes in this study, it has been well documented by previous studies that the population with CAI has less weight-bearing dorsiflexion. To restore a certain amount of functional movement, purposefully targeting dorsiflexion through stretching the tissues surrounding the talocrural joint has been popular among athletes in the past few decades. Therefore, dorsiflexion deficits may not be present in elite athletes who stretch regularly.

It was hypothesized that a significantly higher prevalence of CAI would be present among the contact and acrobatic sports groups. However, in our study no sports team showed a higher prevalence. Surprisingly, we found that diving and trampolining, the acrobatic sports, had the highest percentage of CAI (75%) (Table 3). We suspect that both of these exercises require high-intensity and highly-rated jumping, in which hyperflexible ankles further increase the risk of CAI. In contrast, fencing (n = 8 out of 23; 34.8%) had a very low incidence of CAI. Fencing generally requires only forward and backward motion but not lateral motion or jumping. As a result, they experience more injuries to hip and knee than ankle [32], which may contribute to the low percentage of CAI. By sports category, although there was no significant difference between the three categories (acrobatic, contact, and non-contact) (X^2^(2) = 4.947, *p* = 0.08) (Table 4), acrobatic athletes had a significantly higher prevalence of CAI than non-contact sports (X^2^(1) = 4.674, *p* = 0.03). Previous research [33] showed that high-risk activities for ankle injury include team sports, as well as sports involving contact or jumping. One study [34] indicated that acrobatic athletes were frequently injured in training, with most of the injuries located in the lower limbs. The high incidence of sprains in acrobatic athletes is probably due to the performances of dynamic skills, falling from height, and missed landings. Therefore, personalized training programs, injury prevention, and jumping strategies for acrobatic athletes should be taken into account.

The CAIT was recommended by the IAC as a validated ankle instability-specific self-reported questionnaire [4]. The Chinese version of CAIT has proved its feasibility, reliability, and validity [35,36]. The CAIT comprises nine items for a total score ranging from 0 to 30. The lower scores of CAIT indicate more severe instability, and a cutoff score of CAIT ≤ 27 was chosen by the developers of CAIT to indicate ankle instability [15]. With a narrow range of CAIT scores indicating a stable ankle, some individuals, such as those recovering from ankle sprain who have a history of ankle injury without subjective symptoms of ankle instability, will also be classified as having an unstable ankle. To exclude the recovered patients, the cutoff score of CAIT was suggested to be ≤24 by several researchers [37,38]. In accordance with the prevalence of CAI, the CAIT value showed no significant difference in sports teams (Figure 4).

A previous study [39] graded lateral ankle sprains as follows: Grade I, no loss of function, no ligamentous laxity; Grade II, some loss of function, positive anterior drawer test and negative talar tilt test; and Grade III, near total loss of function, positive anterior drawer and positive talar tilt tests. To our knowledge, no previous studies investigated CAI using the anterior drawer test and talar tilt test to confirm the severity of CAI. The results show that a proportion of CAI athletes had positive anterior drawer and positive talar tilt tests, which indicated anterior talofibular and calcaneofibular ligament problems [40,41]. However, the tests were generally used for acute lateral ligament injuries in early stages, which only indicate ligamentous integrity [42]. It is difficult to determine whether athletes’ ankles are unstable just based on positive signs in anterior drawer or talar tilt tests. Future studies could focus on the clinical examination of CAI patients to obtain more quantitative descriptive data and feature information.

Our study has several limitations. Firstly, although our study diagnosed CAI based on the IAC criteria, the effect of recall bias is difficult to avoid. Some research suggests that the IAC criteria cannot capture the entire population of CAI. For example, some athletes with CAI are excluded because they have a history of knee surgery, which leads to a misestimation in the prevalence of CAI. Secondly, the number of athletes in each gender was limited and unequal. If the number of the female athletes had been the same as that of the male athletes, the prevalence of CAI in elite athletes might have been different. In consideration of the statistical significance determined in the study, we hope that the data can be useful for further research, even in a limited sample size. Finally, the performance of athletes was not counted in this study, which may have a correlation with the prevalence of CAI. A prospective follow-up study on the changes in athletes’ performance after the first-time ankle sprain and during the development of CAI should be conducted to overcome this limitation.

## 5. Conclusions

The aim of our study was to identify the prevalence of CAI in elite athletes. Our study found that 85.9% of athletes (170/198) had previous ankle sprains, and the prevalence of CAI was 64.6% (128/198 athletes). Female athletes had a significantly higher prevalence of CAI than male athletes. Prevention measures of CAI should be a focus for acrobatic athletes, as they have a higher risk of developing CAI. When applying ankle protections for athletes, gender and sports categories should be taken into consideration.

## Figures and Tables

**Figure 1 jcm-11-07478-f001:**
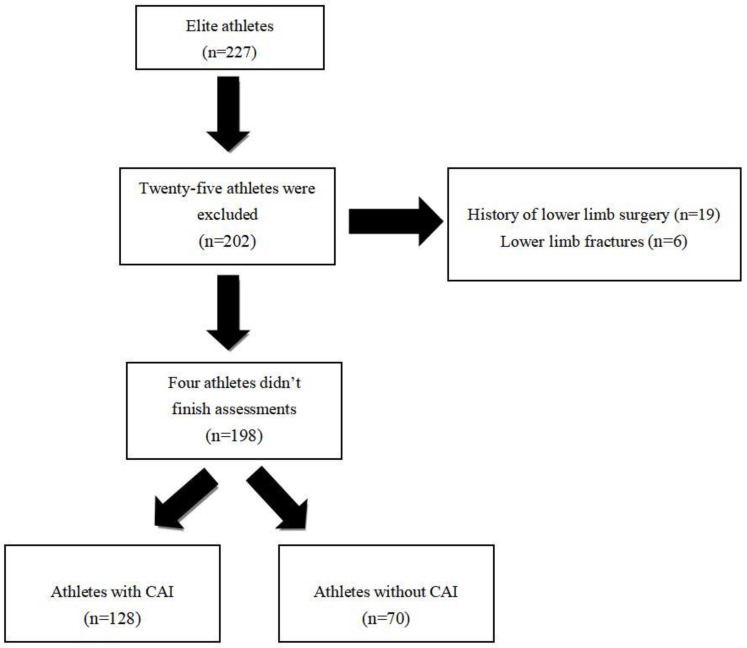
Participants inclusion and exclusion process.

**Figure 2 jcm-11-07478-f002:**
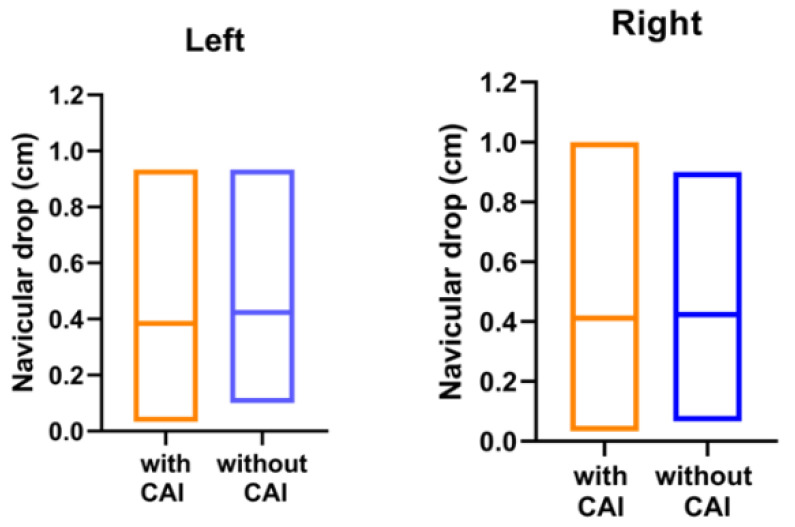
Navicular drop between the groups with and without CAI (measured using navicular drop test). There is no significant difference between the groups (left: *p* = 0.12; right: *p* = 0.68).

**Figure 3 jcm-11-07478-f003:**
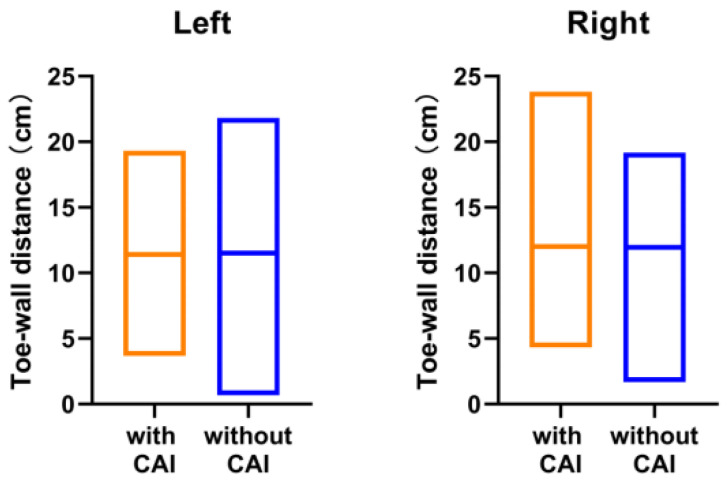
Dorsiflexion ranges between the groups with CAI and without CAI (measured using dorsiflexion lunge test). There is no significant difference between the groups (left: *p* = 0.82; right: *p* = 0.88).

**Figure 4 jcm-11-07478-f004:**
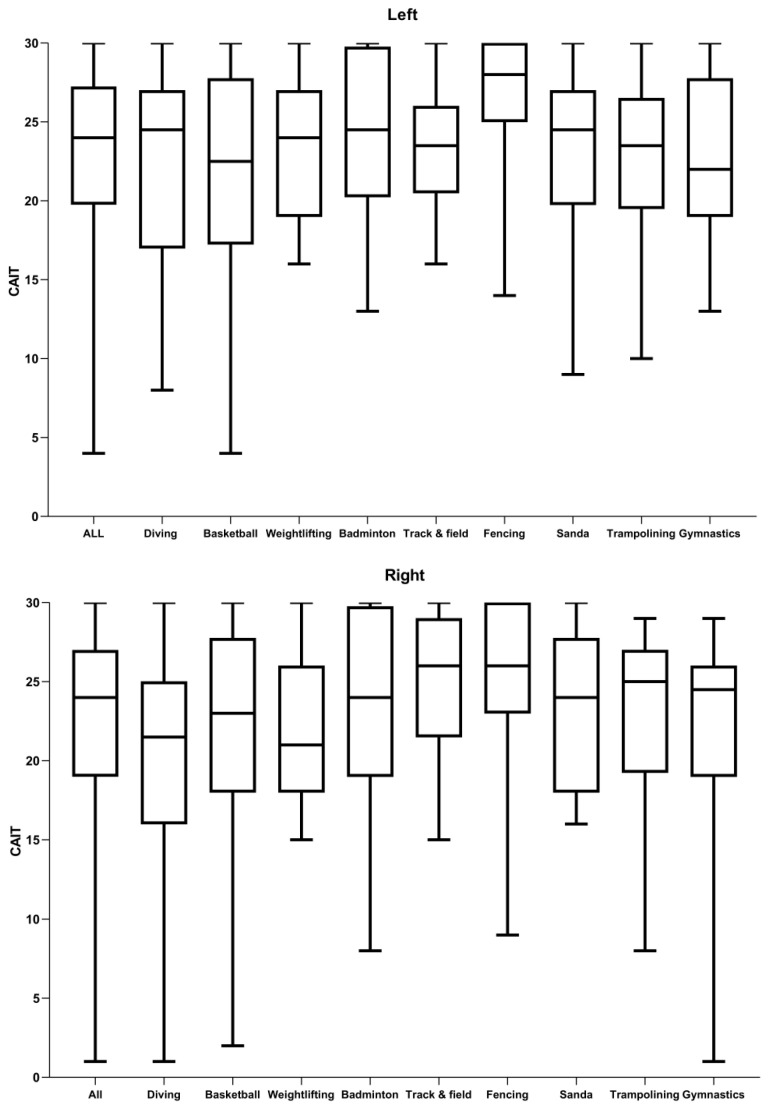
The CAIT value for each sports team.

**Table 1 jcm-11-07478-t001:** Demographic differences between participants with and without chronic ankle instability (CAI).

	CAI (n = 128)	without CAI (n = 70)	
	Mean (SD)	Mean (SD)	*p*
Age (year)	21.33 (3.91)	20.97 ± 3.76	0.584
Height (cm)	172.72 (15.73)	175.24 ± 15.22	0.966
Weight (kg)	65.55 (17.45)	67.36 ± 15.76	0.636
BMI (kg/m^2^)	21.63 (3.24)	21.75 (3.44)	0.806
CAIT score			
Left ankle	20.78 (5.27)	27.80 (1.83)	<0.001
Right ankle	19.64 (5.93)	27.80 (1.82)	<0.001

CAI: chronic ankle instability, SD: standard deviation, BMI: body mass index, CAIT: Cumberland Ankle Instability Tool.

**Table 2 jcm-11-07478-t002:** History of ankle sprains and prevalence of CAI between genders.

	Female (n = 69)	Male (n = 129)	
	n (%)	n (%)	*p*
Ankle sprain history			
Previous ankle sprain	62 (89.86)	108 (83.72)	0.238
No ankle sprain	7 (10.14)	21 (16.28)	
CAI prevalence			
No CAI	15 (21.74)	55 (42.64)	0.010
Unilateral CAI	19 (27.54)	31 (24.03)	
Bilateral CAI	35 (50.72)	43 (33.33)	

CAI: chronic ankle instability (meeting the IAC criteria, including the CAIT cut-off value in those with previous ankle sprains).

**Table 3 jcm-11-07478-t003:** Ankle sprain history and prevalence of chronic ankle instability (CAI) in different sports.

Sport	Total	Ankle Sprain History		CAI Prevalence		
		Previous Ankle Sprain	No Ankle Sprain	No CAI	Unilateral CAI	Bilateral CAI
	n	n (%)	n (%)	n (%)	n (%)	n (%)
All	198	170 (85.9)	28 (14.1)	70 (35.4)	50 (25.3)	78 (39.4)
Diving	24	22 (91.7)	2 (8.3)	6 (25.0)	8 (33.3)	10 (41.7)
Basketball	40	34 (85.0)	6 (15.0)	13 (32.5)	9 (22.5)	18 (45.0)
Weightlifting	19	16 (84.2)	3 (15.8)	6 (31.6)	3 (15.8)	10 (52.6)
Badminton	20	15 (75.0)	5 (25.0)	7 (35.0)	5 (25.0)	8 (40.0)
Track and field	20	17 (85.0)	3 (15.0)	8 (40.0)	5 (25.0)	7 (35.0)
Fencing	23	17 (73.9)	6 (26.1)	15 (65.2)	3 (13.0)	5 (21.7)
*Sanda*	12	10 (83.3)	2 (16.7)	4 (33.3)	3 (25.0)	5 (41.7)
Trampolining	20	19 (95.0)	1 (5.0)	5 (25.0)	8 (40.0)	7 (35.0)
Gymnastics	20	20 (100.0)	0 (0.0)	6 (30.0)	6 (30.0)	8 (40.0)

**Table 4 jcm-11-07478-t004:** Prevalence of chronic ankle instability (CAI) for each category of sport.

Sport	Total	Acrobatics (n = 54)		Contact (n = 52)		Non-Contact (n = 82)		Differences between Groups
		without CAI	with CAI	without CAI	with CAI	without CAI	with CAI	
	n	n (%)	n (%)	n (%)	n (%)	n (%)	n (%)	
All	198	17 (26.6%)	47 (73.4%)	17 (32.7 %)	35 (67.3%)	36 (43.9%)	46 (56.1%)	X^2^(2) = 4.947, *p* = 0.08
Female	69	2 (7.1%)	26 (92.9%)	3 (23.1%)	10 (76.9%)	10 (35.7%)	18 (64.3%)	X^2^(2) = 6.734, *p* = 0.03
Male	129	15 (41.7%)	21 (58.3%)	14 (35.9%)	25 (64.1%)	26 (48.1%)	28 (51.9%)	X^2^(2) = 1.409, *p* = 0.49

**Table 5 jcm-11-07478-t005:** The positive rate of anterior drawer tests and talar tilt tests for CAI athletes.

Sport	Total	Left Ankle			Right Ankle		
		CAI	ADT(+) in CAI	TTT(+) in CAI	CAI	ADT(+) in CAI	TTT(+) in CAI
	n	n (%)	n (%)	n (%)	n (%)	n (%)	n (%)
All	198	100 (50.5)	72 (72.0)	46 (46.0)	106 (53.5)	79 (74.5)	45 (42.5)
Diving	24	12 (50.0)	8 (66.7)	8 (66.7)	16 (66.7)	13 (81.3)	4 (25.0)
Basketball	40	21 (52.5)	17 (81.0)	12 (57.1)	24 (60.0)	15 (62.5)	9 (37.5)
Weightlifting	19	10 (52.6)	6 (60.0)	4 (40.0)	13 (68.4)	10 (76.9)	6 (46.2)
Badminton	20	10 (50.0)	8 (80.0)	7 (70.0)	11 (55.0)	7 (63.6)	5 (45.5)
Track and field	20	11 (55.0)	10 (90.9)	4 (36.4)	8 (40.0)	7 (87.5)	3 (37.5)
Fencing	23	5 (21.7)	5 (100.0)	3 (60.0)	8 (34.8)	8 (100.0)	4 (50.0)
*Sanda*	12	6 (50.0)	4 (66.7)	2 (33.3)	7 (58.3)	6 (85.7)	4 (57.1)
Trampolining	20	13 (65.0)	8 (61.5)	4 (30.8)	9 (45.0)	9 (100.0)	6 (66.7)
Gymnastics	20	12 (60.0)	6 (50.0)	2 (16.7)	10 (50.0)	4 (40.0)	4 (40.0)

ADT: anterior drawer test; TTT: talar tilt test.

## Data Availability

Not applicable.

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
