# Peer review of "The Prevalence and Characteristics of Chronic Ankle Instability in Elite Athletes of Different Sports: A Cross-Sectional Study"

_jcm, 2022, doi:10.3390/jcm11247478_

Round 1
Reviewer 1 Report
The topic of the presented manuscript is important, relevant, and original. This is a well written manuscript but some issues could be improved.
Introduction seems enough but could achieve a deeper knowledge about state of art.
The research hypothesis are not described
The chronology is missing.
Discussion section seems confused, needs to be rewritten in some paragraphs.
Reviewer 2 Report
TITLE: Does the title clearly portray the subject and purpose of the study?
It does come in no issues with the title.
ABSTRACT: Doers the abstract accurately reflect the study? Are all pertinent finds included?
The abstract accurately reflects the study, and all relevant components are included. However, the abstract highlights the need for excessive English language edits. An English reader will understand the need for revision of the following lines, which do not read appropriately:
Line 10
Line 23
Line 30
INTRODUCTION: What is the authors' original research question, and does their study support or fulfill it?
The authors aimed to assess the prevalence of chronic ankle instability in an athletic population at their university. In so doing, this was a cross-sectional analysis.
Unfortunately, while this is an interesting study idea, the manuscript needs significant revision. Not just for English language, but for simple edits such as the following:
“Ankle sprain is one of the most common injuries in the active population, and they 29 were occurred in competitive sports”
“CAI is defined as a patient (1) who wasremoved from the initial ankle sprain over a year”
“Therefore, to classify patients with CAI, it should be considering to use a combination of self-reported questionnaires, clinical assessment and objective measurement to assess the stability of primary ligaments in ankle”
This is not a comprehensive listing of the English language edits that need to be made. Otherwise, the introduction is fine in my opinion.
METHODS: Was the research method or study design appropriate? Is it presented sufficiently so that other researchers can duplicate them? Are the sample sizes adequate? Are the statistical analyses appropriate and correct?
The study design is appropriate for a cross-sectional analysis. The methods are comprehensive in their nature, although English language edits need to be made in line with the rest of the manuscript. The sample size does appear to be adequate, and the statistical analysis are appropriate, using Mann Whitney U for nonparametric data and t test for parametric.
RESULTS: Do the results answer the original research questions, as demonstrated in the Results section and tables and figures?
The results take a stab at answering the original research question, but many sports are not represented. For example, soccer, one of the most popular sports in the world, was not represented in this study. It is probably unfair to say as a sweeping generalization that athletes have such and such a prevalence of CAI based on this study.
Furthermore, and perhaps most concerning, the numbers obtained by these authors differ vastly from previous reports.
This study, with the link below, reports a rate of 20%, which is threefold lower than the rate presented by these authors, in a sample over 1000, which is a much more robust sample than the one presented here, less than 200. Importantly, while the sample here is roughly 200, the sample for each individual sport is a lot lower, and they are probably not powered enough to detect intersport differences, as I would imagine, for example, a weightlifter would have a different rate of CAI from a basketball player.
https://pubmed.ncbi.nlm.nih.gov/32118082/#:~:text=Results%3A%20The%20overall%20prevalence%20of,PedsQL%3A%2093.5%20%C2%B1%209.1).
Maybe I'm missing something, but these results seem extraordinarily inaccurate, and just simply not on par with what makes sense from a clinical standpoint or from the prior literature. If well over half of patients have chronic ankle instability, then clearly we need to adjust our metric for what we define as ankle instability…
DISCUSSION: Is the Discussion balanced? Does it put the results in context? Do the authors acknowledge the limitations of the study?
The authors take a stab at why their numbers are so high, but I think their reasoning for why they report such high rates are inadequate.
I appreciate their use of the CAIT and the definition of CAI from the international angle consortium. However, their results just differ too strongly from other studies which have used these same definitions and scoring metrics.
Consider the study below, which has a far lower prevalence than the one they present here. Are we sure there wasn't some measurement error? We need to be sure we hold the high standard for publication and we do not publish erroneous results, these results just being so far off everything else that's been reported are concerning.
https://pubmed.ncbi.nlm.nih.gov/32949959/
CONCLUSIONS: Are the conclusions supported by the study findings? Does the study provide new, unique, or confirmatory findings? Will the findings be of interest to clinicians or to the public?
The conclusions are supported by the findings of the study, but see my above comments or concerns regarding these findings.
TABLES AND FIGURES: Are all data presented in the text and tables and figures consistent? Do the tables clearly present information not easily summarized in the text of the paper? Are all of the tables necessary? Are the figures necessary and appropriate? Are they of high quality and clearly labeled? Can any be deleted?
REFERENCES: Is the References section complete, or is it excessive? Does it include all of the necessary current, relevant sources
References and tables are fine.
Round 2
Reviewer 2 Report
I appreciate the authors careful responses to my comments. While the authors have done an excellent job at convincing me that their reported rates of chronic ankle instability are not necessarily out of bounds, given some of the reports in other more elite athlete sample sizes, unfortunately, the English language deficits are just too substantial to warrant publication.
An English language Ryder will read the following paragraph and recognize all the issues therein:
Ankle sprain is one of the most common musculoskeletal injuries, with a high inci- 28 dence in competitive sports[1, 2]. Acute ankle sprain has a high recurrence rate, which 29 mainly leads to develop chronic ankle instability (CAI). CAI is characterized by a patient 30 (1) who was suffered from the very first ankle sprain over a year; (2) exhibiting a propen- 31 sity for recurrent ankle sprains; (3) experienced frequent perceptions of the ankle giving 32 way; (4) suffered from repetitive pain, swelling, diminished ROM, weakness, and reduced 33 daily function[3, 4]. The reoccurring harm, restricted movement and imbalancing caused 34 by CAI lead the individuals prone to further injuries, which significantly affect the ath- 35 letes’ career expectation for practices and competitions
likewise, this paragraph:
The primary aim of our study was to provide an argumentative evidence of the prevalence and characteristics of CAI. Considering that most of the CAI articles were investigated into general population, we aimed to discover whether there are CAI problems among elite athletes without focusing on identifying the risk factors of CAI. In our study, the prevalence of CAI was 64.6%(128/198)(Table 1). Previous studies reported the preva lence of perceived ankle instability was between 20% and 47%[18-20]. The prevalence of CAI vary dramatically in specific groups of subjects. The high prevalence of CAI in elite athletes might be caused by: (1) the participants in this study were playing in a higher professional league and compete at a higher intensity, as well as having a higher rate of previous ankle injures; (2) elite athletes might place more hyperflexible pressure on their ankles than general athletes or amateur athletes, especially in the acrobatics sports; (3) most of elite athletes started highly specialized training since kid, which makes them ex- 196 pose to a higher risk of overuse injury. Future study is required to fully investigate the relevance between the professional level of athlete and prevalence of CAI.
I think an English language service needs to be hired or something like that. I would be happy to re-review
However, the content has been greatly improved, and they have done a great job at explaining their high reported rates of CAI.
